# Estimating the Psychological Harm Consequence of Bullying Victimization: A Meta-Analytic Review for Forensic Evaluation

**DOI:** 10.3390/ijerph192113852

**Published:** 2022-10-25

**Authors:** Álvaro Montes, Jéssica Sanmarco, Mercedes Novo, Blanca Cea, Ramón Arce

**Affiliations:** Unidad de Psicología Forense, Facultad de Psicología, Universidad de Santiago de Compostela, 15782 Santiago de Compostela, Spain

**Keywords:** mental harm, restitution, compensation, victim of crimes, forensic psychological evaluation

## Abstract

The prevalence of traditional bullying victimization has been estimated at around 36%, while that of cyberbullying has been estimated at 15%. The victimization of bullying brings with it harm to mental health that must be compensated for, after a forensic evaluation, by the aggressor or legal guardian. Thus, a meta-analytic review was undertaken with the aim of knowing the effect of bullying victimization on psychological harm, as well as quantifying the magnitude of the harm and estimating the probability that no harm associated with bullying victimization is generated. Method: A random-effects correlational meta-analysis correcting effect size by sampling error and criterion and predictor unreliability was performed. Results: The results exhibited a positive (i.e., more victimization and more psychological harm) and significant mean true effect size, implying an average psychological harm associated to bullying victimization of 29.7%. Nevertheless, 26.7% of students victimized by bullying did not develop psychological harm. Conclusions: Bullying victimization causes psychological harm, with an average increase in psychological harm associated with bullying victimization of 29.7%.

## 1. Introduction

Violence in the school context has long been the subject of academic, institutional, social, legal, and health interest because of the associated negative outcomes. Violence in scholar setting includes physical fights (students decide to fight each other), physical attacks (a student or students hit or hurt another student), sexual violence (attempted or completed nonconsensual sex acts, abusive sexual contact, and noncontact sexual abuse), and psychological violence (verbal abuse, emotional abuse, coercion, and social exclusion). In the 1970s, Dan Olweus called bullying a configuration of school violence and defined it as “a student is being bullied or victimized when he or she is exposed, repeatedly and over time, to negative actions on the part of one or more other students” [1] (p. 1173). Over time, this definition was specified with four diagnostic criteria that facilitates not only the diagnosis of bullying, but also the differential diagnosis with other types of aggressive behaviors that occur between equals such as school violence, conflicts, physical fights, peer victimization, school phobia, sexual abuse, or teasing [2,3,4,5,6,7].

The first criterion (harmful behaviors or strategies) establishes that there must be aggressive behaviors or strategies that have the objective of physically or psychologically injuring someone; they can be physical, verbal, or relational behaviors [8]. The second requires that the behaviors must be directed with the intention of causing harm to the harassed person (intentionality criterion). These first two criteria can be combined into one to give rise to the criterion of “intention to cause harm” [4,9]. The third determines that there must be an asymmetry of power between aggressor and victim (power imbalance criterion) [9,10]. The fourth criterion implies that these harassment behaviors must have a repeated character over time (criterion of periodicity and chronicity); that is, the actions must be repeated and prolonged over time to be considered bullying [11,12]. Given that these criteria are also met in so-called cyberbullying, it is then a form of bullying in which students use electronic devices or in cyber context [4,13,14]. Additionally, for the purposes of a forensic evaluation, a fifth criterion is necessary [15]: victimization. In this regard, [16] the Declaration of Basic Principles of Justice for Victims of Crime and Abuse of Power (hereafter, the UN Declaration of Basic Principles of Justice for Victims) states that victims of a crime and abuse of power are “persons who, individually or collectively, have suffered harm, including physical or mental injury, emotional suffering, economic loss, or substantial impairment of their fundamental rights, through acts or omissions that are in violation of criminal laws operative within Member States, including those laws proscribing criminal abuse of power” (Article 1). Thus, the UN Declaration of Basic Principles of Justice for Victims requires that the person has been the target of acts or omissions that infringe criminal law and that, as a consequence, has suffered harm, including physical injury, psychological injury (i.e., mental injury or emotional suffering), economic loss, or substantial impairment of their fundamental rights. In consequence, if acts or omissions do not infringe criminal law or do not produce harm, there is no victim. Nevertheless, in children, the potential harm as a consequence of abuse or maltreatment is victimization [17]. In fact, psychological harm may be developed through delayed onset [18,19] or with delayed expression [20]. However, victimization through child abuse may not lead to psychological harm [21]. Thus, in children, bullying victimization does not invariably require that psychological injury be probed. Nonetheless, psychological harm is key evidence supporting a case as it endows the status of victim to the claimant and confers credibility to their testimony [22].

Posttraumatic stress disorder (PTSD) is the forensic psychological evidence of psychological harm [23,24,25,26]. This is because the symptoms of the PTSD must be associated with a traumatic event [20]. Additionally, comorbidity surveys reported that PTSD is systematically associated with victimization [27,28]. If criterion A (traumatic event: exposure to actual or threatened death, serious injury, or sexual violence) of the PTSD is not met (the stressor does not have the severity or is of a different type than listed in PTSD criterion A, e.g., psychological violence) but the remaining criteria for PTSD are met, then the diagnosis is an adjustment disorder [20]. Thus, adjustment disorder associated with bullying should also be diagnosed as psychological harm. PTSD is highly comorbid and multi-comorbid, being inherently associated with depression and anxiety disorders [29]. Varying with the nature of the traumatic event or stressor, many disorders may be associated with PTSD. In childhood abuse, PTSD is linked to the risk of suicidal ideation and suicide behaviors including completed suicide [20,30,31]. In any case, the diagnosis of other disorders without the simultaneous existence of PTSD does not constitute forensic evidence of victimization as the relationship between the injury sustained and traumatic event or stressor has not been established [32].

The potential mental health harms associated with bullying and its high prevalence, (0.360 in traditional context and 0.156 in cyber context [13]) led us to design a meta-analytic review with the aim of finding out the effect of bullying victimization on psychological harm (i.e., posttraumatic stress disorder and adjustment disorder), as well as quantifying the magnitude of harm (bullies should, where appropriate, make fair restitution to victims; see article 8 of the UN Declaration of Basic Principles of Justice for Victims), and estimating the probability of no harm associated with bullying victimization (in children, potential harm is victimization [17]).

## 2. Method

### 2.1. Literature Search

A sensitive multisource search involving four different meta-search strategies was performed: Google Scholar; reference databases of scientific quality evaluation (i.e., Web of Science and Scopus); specialized reference databases (i.e., PsycInfo, Dialnet, TESEO, and Psicodoc); a review of the bibliographic references of the papers that were selected. The initial search was made on the basis of the following broad search terms and commands: school bullying AND psychological harm. In line with a successive approximation method, the keywords of the selected articles were revised in search of narrow terms.

### 2.2. Inclusion and Exclusion Criteria

To the selected articles, the following inclusion criteria were applied: (a) that the evaluation of bullying was carried out with a validated psychometric instrument; (b) that the bullying referred to the school context; (c) that the forensic victimization of harm (PTSD, Adjustment Disorder or Acute Stress Disorder) was evaluated; (d) that the assessment of psychological harm was with a validated psychometric instrument; (e) that the effect size of the relationship between bullying victimization and psychological harm was provided or sufficient data to calculate it were available.

The following exclusion criteria were applied: (a) that the evaluation of bullying was not in school context (e.g., workplace bullying); (b) that the bullying measure did not guarantee a differential diagnosis of other similar victimizations (e.g., school violence, conflicts, physical fights, peer victimization, school phobia, and teasing); (c) that the measure of psychological harm involved other disorders that did not allow establishing a causal relationship between bullying victimization and harm to mental health (e.g., depression and anxiety); (d) that the measurement of psychological harm was not a reliable and valid instrument.

As a result of this search and the application of the inclusion and exclusion criteria, 10 primary studies were selected (see flow diagram in Figure 1). All of them were journal articles, obtaining a total of 10 effect sizes (correlations) and a cumulative sample of 9030 subjects (see Table 1).

### 2.3. Coding of Primary Studies

Coding was performed by two independently trained researchers (between-rater concordance) in the following categories of analysis: (a) study reference; (b) sample size; (c) identification of the scales and their reliability; (d) registration or calculation of the size of the effect. After 1 week, the same researchers coded half of the studies (within-rater concordance). The results of the between- and within-rater true concordance ($\bar{\kappa }$ [43], which corrects the Cohen’s kappa verifying the exact correspondence of coding (true kappa), was perfect ($\bar{\kappa }$ = 1).

Furthermore, one of the raters was consistent with another rater in another study [44] i.e., between-context concordance. Thus, any other trained rater would find the same dataset and that coding perfectly reflects the analysis categories. A successive approach procedure was applied to the primary studies [45], constituting two researchers with experience in forensic evaluation who separately identified the moderators. Then, raters discussed and reached consensus of the moderators. The following moderators were identified: time of the PTSD measure (actual vs. delayed); type of bullying (traditional bullying or face-to-face bullying and cyberbullying); population (primary school students, secondary school students, or university students).

### 2.4. Data Analysis

A correlational meta-analysis was performed correcting effect size for sampling error, as well as predictor and criterion unreliability [46]. Effect sizes were taken directly from the primary studies or transformed to *r* from other effect sizes (e.g., odds ratio and Cohen’s *d*) or from common statistics (e.g., χ^2^, *F*, and *t*). Specifically, in the present meta-analysis, the following metrics were computed: the sample size weighted mean observed correlation ($\bar{r}$); sample size weighted observed standard deviations of correlations (*SD*r); mean true score correlation (ρ); standard deviation of true score correlations (*SD*ρ); percent variance in ρ correlations attributable to all artifacts (%VAR); 95% confidence interval for *r* (95% CI); the 80% credibility interval for ρ (80% CI). When the 95% CI of *r* does not include zero, the average effect size is significant. Nonetheless, trivial effects, as a consequence of large *N*s, may result in significance [46]. A trivial effect for *r* is 0.05; then, if the 95% confidence interval for *r* passes for 0.05, the effect is trivial [47]. If the lower limit of the 80% credibility interval does not include zero, it means that the result is generalizable to 90% of the potential studies. If, in addition, the lower limit is greater than a trivial effect, the results are not only generalizable, but the minimum effect to be expected is also significant and nontrivial [48]. If artifactual errors (%VAR) explains the bulk of the variance (>75%; 75% rule [49]), then the non-explained variance is not systematic, describing the mean true correlation between bullying victimization and psychological harm. Conversely, if the variance explained by artefacts is less than 75%, then moderators of the effect exist and should be studied. The mean true effect size was qualitatively interpreted according to Cohen’s categories [50] as small (ρ = 0.10), moderate (ρ = 0.30), and large (ρ = 0.50) and quantitatively in terms of the probability of superiority of the effect size (PEES [44]), an estimation of the probability of the observed effect size above all possible effect sizes. The magnitude of the psychological harm associated with bullying victimization was measured as BESD [51], and the increase over a trivial effect (harm) was measured in terms of the effect incremental index (EII [52]). Lastly, the statistical model error, i.e., an evaluation of the probability of non-harm associated with bullying victimization was estimated with the Probability of an Inferiority Score (PIS [49]).

### 2.5. Predictor and Criterion Reliability

The effect size was corrected by three sources of artifactual variance: sampling error, predictor unreliability, and criterion unreliability. The reliability of the predictor (bullying victimization) and the criterion (PTSD measure) was obtained from the primary study and, when not reported, it was calculated either from the instrument manual or from the instrument creation and validation study. As a result, an average reliability coefficient of 0.811 [0.756, 0.866] was obtained for the bullying victimization instruments and 0.895 [0.866, 0.924] for PTSD symptoms. Effect sizes could not be corrected for range restriction because the data needed for this estimation were not reported in the primary studies and could not be obtained by other means. Table 1 summarizes the predictor and criterion reliability.

## 3. Results

### 3.1. Analysis of Atypical Values

The data were explored in search of extreme values (±3 × IQR), outliers (±1.5 × IQR), and abnormal values with the application of Chauvenet’s criterion (±1.96 × *SD*). No extreme, outlier, or abnormal effect sizes were observed. Thus, all primary data were normal. Additionally, data had a normal distribution, W(10) = 0.893 (ns).

### 3.2. Meta-Analysis for the Relationship between Psychological Harm Consequence and Bullying Victimization

The results of the meta-analysis (see Table 2) for estimating the relationship between bullying victimization and psychological harm, with a total sample of 9030 individuals and 10 effect sizes, revealed a significant true average effect size (ρ) (the 95% confidence interval for *r* did not include zero), positivity (the greater the bullying victimization, the more psychological harm, i.e., symptoms of posttraumatic stress disorder), generalizability (the 80% confidence interval did not include zero), and a moderate magnitude (ρ ≈ 0.3; above 67.0% of all positives, PSES = 0.670). From these results, it can be deduced that the victimization of behaviors constituting bullying explained 8.8% of PTSD (ρ^2^ i.e., bullying victimization explains the 8.8% of PTSD, while other causes account for the remaining variance—multicausality), that the average increase in psychological harm associated with bullying victimization was 29.7%, and that the increase in the probability of harm on a trivial effect (0.05) was 83.1% (EII = 0.831). In short, the victimization of bullying behaviors produced psychological harm that was susceptible to forensic evaluation and, hence, to judicial demonstration. However, the percentage of variance explained by the artefactual errors was <75%; thus, the results were influenced by moderators. On the contrary, 26.7% (PIS = 0.267) of the students victimized from bullying did not experience psychological harm (error of the statistical model); that is, around one-quarter of students victimized with bullying did not develop psychological harm (rape victims were around 45% [53]). An effect moderator study could not be carried out because k (<3) and/or N (<300) were insufficient.

## 4. Discussion

The results of this meta-analysis are subject to generalization limitations that must be kept in mind. First, correlational meta-analyses do not control for the effects of other causes. Thus, the effect is contaminated (multicausality) in part by other stressors. Nevertheless, the experimental designs guarantee that the bulk of the variance (causality) is explained by bullying victimization. In addition, the increase in the effect of a trivial effect (effect attributed by chance to other causes) is greater than 80%; that is, it was estimated that more than 80% of chance is caused by bullying victimization. Second, the variability is explained by moderators that have not been sufficiently studied; thus, their effects are unknown. Taking into account the limitations, the results of this meta-analysis suggest that bullying victimization is associated with psychological harm. Consequently, the forensic test of psychological harm must be carried out systematically in all cases of reported bullying. However, not all victimization leads to harm (the estimated probability was approximately one in four). The results support that verification of psychological harm is not a strict criterion for bullying victimization. Criminal victimization entails, according to the restitution principle (Article 8) of the UN Declaration of Basic Principles of Justice for Victims, the payment for the harm by the offenders or third parties responsible for their behavior (in bullying cases, by parents or legal guardians). As an average, the psychological harm consequence of bullying (~30%), translated to the Global Assessment Functioning Scale (GAF [54]), would be moderate (50—60 [range for moderate]) in contrast to healthy individuals (81–100 [range for transient or asymptomatic]). As for the population with psychological harm, the lower harm expected (lower limit of the credibility interval for the psychological harm) would be around 20%, while the extreme harm (upper limit of the credibility interval for the psychological harm) would be about 40% (serious/severe harm in GAF Scale), requiring treatment and attention (i.e., suicidal ideation and suicide behaviors). Harm under or over these limits, although possible, would be abnormal. In relation to no harm registered victims (in children, potential harm consequence of abuse or maltreatment is victimization), harm compensations (delayed onset or expression) could be quantified as average (~30% i.e., moderate harm).

## 5. Conclusions

In light of the harm caused and the high prevalence of cases [13], it is necessary to implement prevention programs that have been shown to be effective in preventing and reducing extreme aggression [55,56] and, for reducing the psychological harm sequelae, submitting the attendant to individual and group cognitive–behavioral therapy [57]. As for the judicial setting, bullies should be sentenced to restitute the psychological harm to victims, while those victimized with bullying that do not present psychological harm should be restituted with the mean observed harm (potential harm in general definition of WHO’s child victims). Nevertheless, students victimized with bullying must be evaluated case by case due to the large observed variability.

## Figures and Tables

**Figure 1 ijerph-19-13852-f001:**
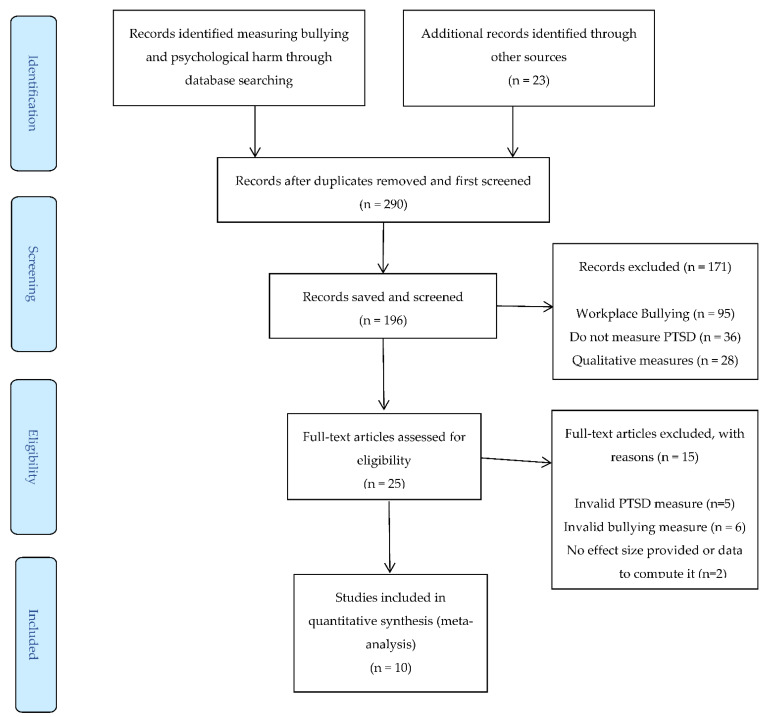
Flow diagram of the meta-analysis.

**Table 1 ijerph-19-13852-t001:** Primary studies data.

Study Cite	*N*	*r* _xy_	Bullying Measure	*r* _xx_	PTSD Measure	*r* _yy_
Andreou et al. (2021) [33]	150	0.23	Olweus Bully/Victim Questionnaire (BVQ)	0.65	PTSD Checklist—Civilian Scale (PCL-C)	0.92
Baldry et al. (2019) [34]	5058	0.19	Olweus Bully/Victim Questionnaire (BVQ)	0.77	Trauma Screening Questionnaire (TSQ)	0.82
Espelage et al. (2016) [35]	482	0.33	University of Illinois Victimization Scale (UIVS)	0.84	Short PTSD Rating Interview (SPRINT)	0.84
Guzzo et al. (2014) [36]	488	0.16	Olweus Anonymous Questionnaire	0.78	Trauma Symptom Checklist—Alternative (TSCC-A)	0.85
Houbre et al. (2006) [37]	162	0.20	Bullying Behavior Scale	0.83	Impact of Event Scale (IES)	0.89
Idsoe et al. (2012) [38]	433	0.34	Roland and Idsoe Scale	0.91	Children Revised Impact of Events Scale (CRIES-8)	0.96
Litman et al. (2015) [39]	358	0.31	Multidimensional Peer-Victimization Scale	0.74	UCLA PTSD Reaction Index	0.91
Manrique et al. (2020) [40]	270	0.36	Adolescent Peer Relations Instrument (APRI)	0.96	UCLA PTSD Reaction Index	0.94
Mateu et al. (2020) [41]	1516	0.36	Olweus Bully/Victim Questionnaire (BVQ)	0.85	Children Revised Impact of Events Scale (CRIES-8)	0.94
Mynard et al. (2000) [42]	113	0.24	Victims Scale	0.78	Impact of Event Scale (IES)	0.88

**Table 2 ijerph-19-13852-t002:** Meta-analytic results.

*k*	*N*	$\bar{r}$	*SD_r_*	ρ	*SD* _ρ_	%Var	95% CI*_r_*	80% CI*_ρ_*
10	9030	0.243	0.0756	0.297	0.0761	24.66	0.196, 0.290	0.212, 0.382

Note: *k* = number of correlations; *N* = total sample size; $\bar{r}$ = sample size weighted mean observed correlation; *SD_r_* = sample size weighted observed standard deviation of correlations; ρ = mean true correlation; *SD*_ρ_ = observed standard deviation of the corrected correlations; %VE = percent variance in corrected correlations attributable to all artifacts (sampling error, predictor unreliability, criterion unreliability); 95% CI*_r_* = 95% confidence interval for *r*; 80% CI*_ρ_* = 80% credibility interval for δ.

## Data Availability

The data presented in this study are available on request from the corresponding author.

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
