# Peer review of "Estimating the Psychological Harm Consequence of Bullying Victimization: A Meta-Analytic Review for Forensic Evaluation"

_ijerph, 2022, doi:10.3390/ijerph192113852_

Round 1
Reviewer 1 Report
This is a very well written article, and it adds to the prevalent issue of bullying. However one minor comment; I suggest expanding the conclusion with the implications and the limitations slightly more detailed.
Author Response
Done. Changes are highlighted in blue.
Reviewer 2 Report
This paper presents the results of an interesting meta-analysis study that aimed a) to reveal the effect of bullying victimization on psychological damage (e.g., PTSD, Adjustment Disorder), b) to quantify the magnitude of the harm and c) estimate the probability of no harm associated with bullying victimization.
I think it is a valuable study which is suitable to be published in your journal in order to help your readers on this important topic.
In order to improve the work, the authors are invited to answer the folloing points.
I wonder whether the authors should include also studies concerning the victims (particularly adolescents and young) of acts or omissions, from the employer’s side, that clearly infringe the low incomes and produce harm to the youth (economic loss, work loss, emotional suffering etc.). It is not clear why authors have excluded this extremely important area of victimization.
Additionally, the authors are invited to give a solid argumentation for the need of this review.
Author Response
R: I wonder whether the authors should include also studies concerning the victims (particularly adolescents and young) of acts or omissions, from the employer’s side, that clearly infringe the low incomes and produce harm to the youth (economic loss, work loss, emotional suffering etc.). It is not clear why authors have excluded this extremely important area of victimization.
That is a good appreciation, but this article refers to school bullying victimization, where the role of the employer and economic or job loss are not applied. Passing you appreciation to school bullying, the counterpart are teacher violence against students and teachers’ omissions of assistance to bullying victims. Nevertheless, there is no scientific evidence about the their effects on bullying victimization.
R: Additionally, the authors are invited to give a solid argumentation for the need of this review.
DONE. Changes are highlighted in blue.
Reviewer 3 Report
I found this manuscript relevant and timely. Thank you for addressing this important topic.
- I found the language confusing when discussing the role of victimization in diagnosed PTSD. For example, in line 203, it is stated that the "victimization of behavior constituting bullying explains 8.8% of PTSand ...(line 209) 26.7% of the students victimized with bullying do not present psychological harm. It is possible that the translation to English resulted in an overall message that bullying victimization does not really contribute that much to PTSD?
- Expand the conclusion, possible by aligning the effects of prevention programs with the information on restitution/
Author Response
- I found the language confusing when discussing the role of victimization in diagnosed PTSD. For example, in line 203, it is stated that the "victimization of behavior constituting bullying explains 8.8% of PTSand ...(line 209) 26.7% of the students victimized with bullying do not present psychological harm. It is possible that the translation to English resulted in an overall message that bullying victimization does not really contribute that much to PTSD?
The results are correct. The first data, 8.8% of the variance, is an interpretation of the true effect size (rho), squaring rho the percentage of the explained variance is obtained. This is moderate (>. 5.9%; Cohen 1988). Conversely, the 26.7% is an estimation of the statistical model error i.e., the probability of victimized without psychological harm. The observed probability is low in comparison with other victimization type as rape victimization where the observed probability of no-harm is around 40%. Information about this was added for non-familiarized readers with these data.
2. Expand the conclusion, possible by aligning the effects of prevention programs with the information on restitution/
DONE (changes are highlighted in blue).
Round 2
Reviewer 3 Report
The edits to the manuscript are satisfactory.
Author Response
Thanks a lot for the revision of style and language. It improved the paper quality. The authors agree with you, and in line with the use Royal Academy of English, that scientific papers must be written in passive voice (APA 7th ed. Stated the no use of passive voice in a ). It is attached the revised version. Changes are highlighted in fuchsia.
